# Understanding Carbon Emissions Reduction in China: Perspectives of Political Mobility

**Zhichao Li [1] and Bojia Liu [2,*]**

1 School of International and Public Affairs, Shanghai Jiao Tong University, Shanghai 200030, China
2 School of Public Administration, South China University of Technology, Guangzhou 510641, China
* Correspondence: 2013040958@ecupl.edu.cn

**Abstract:** Climate change is one of the largest challenges facing mankind, and the question of how to reduce carbon emissions has raised extensive concern all over the world. However, due to the lack of mechanisms to explain the impact of political factors on environmental regulatory tools, the evaluation of carbon emissions reduction is insufficient in the majority of previous studies. How to better explore the path of carbon emissions reduction has become the key for China to achieve carbon neutralization as soon as possible. Based on a quasi-natural experiment regarding China's carbon emission trading policy, this paper adopts a difference-in-differences model to address the impact of political mobility on China's carbon emissions trading policy, and the selected pilot and non-pilot provinces of this policy in China enabled the model to be matched. Using a panel database with 30 provincial administrative units as the observation objects, the results show that China's carbon emissions trading policy and the horizontal mobility experience of the provincial governors exert a significant positive effect on carbon emission reduction. Additionally, this study identifies a latent factor previously ignored by the existing literature: the correlation between political factors and carbon emissions. This verifies our theoretical hypothesis that officials transferred from the provincial level tend to have higher performance regarding carbon emission reduction. This paper also provides suggestions for the central government to further plan and implement carbon emission reduction policies and mobilize the incentives of local officials in environmental governance.

**Keywords:** carbon emissions trading; political mobility; difference-in-differences model; policy evaluation





## 1. Introduction

Human survival and sustainable development face severe challenges. Climate change threatens the lives and environmental conditions of humans worldwide and has become a global concern [1]. Increasing carbon emissions have gradually become the main reason for global climate change. According to the annual report by the United Nations Environment Program, global carbon dioxide emissions showed an upward trend [2]. Correspondingly, to achieve green growth, nations all over the world have enacted a number of environmental laws and emission reduction plans. Therefore, reducing carbon emissions is essential for maintaining a healthy ecological ecosystem.

As the largest source of carbon emissions in the world, China has responded with a series of policies such as instituting emission reduction measures and promoting the establishment of an international carbon market to mitigate global warming [3]. To achieve low-carbon development, China issued the carbon emission reduction plan as guidance for different industrial sectors in terms of achieving the reduction targets in 2012. In 2015, the Chinese central government set the policy target that China's carbon emissions will achieve the peak of domestic carbon emissions by 2030. The Chinese government widened its objectives in 2020 by pledging to achieve carbon neutrality by 2060. To meet these climate targets while trying to pursue economic development, China has implemented a carbon emissions trading system [4]. Since the 11th Five-Year Plan, Chinese central government has

added environmental performance, especially carbon emissions, into the local government performance appraisal, and during the 13th Five-Year Plan it was proposed to establish a national carbon emission trading market by 2020.

Previous studies on carbon emission trading policies have explored the factors that influence carbon emission performance, such as technology development [5], total energy consumption [6] and energy consumption structure [7]. Studies on the impact of carbon emission trading policies mainly focused on two categories, which contains sectors and environmental governance. For the sectoral impact, existing studies have mainly focused on the power sector and transport sector [8,9]. However, it can be found that the existing empirical research was limited to certain sectors related to carbon trading policies while ignoring interactions with other perspectives, or was concentrated on macro-economic simulation instead of specific environmental activities and behaviors based on different perspectives.

Regarding the overall impact on a national level, research on the driving mechanism of China's carbon emissions reduction is relatively lacking and has insufficient evidence. Previous studies ignored the role of government officials in carbon emission reduction. Few studies chose to use factors such as government performance and human behavior to explore an in-depth explanation of China's carbon emissions reduction, especially the impact of political leaders on environmental governance. The literature shows that in a political system, the attention distribution of political leaders may affect or even change policy objectives, their implementation methods and performance [10]. The most persuasive theory about China's economic growth miracle is the "Promotion Tournament Model Theory" [11], which points out that the incentive of local officials is a fundamental motivation of China's economic development. To improve economic performance, Chinese central government has established the promotion tournament incentive structure, which has motived local officials with GDP growth as the core indicator. Understanding the influence of local officials on carbon emissions is very important for policy implication. However, considering the political tournament mechanism in China, local officials face two kinds of promotion incentives, including economic development and emissions reduction: can the implementation of policies on carbon emission trading effectively promote the performance of carbon emission reduction?

In this study, we explored whether the current carbon emissions trading policy can promote emission reduction performance, and we proposed a theoretical framework that explains the relations among political mobility, tenure, and carbon emission performance. In areas with similar natural resources, human capital and technological innovation capacity, local carbon emission performance may be different due to the different political experiences of leaders. By understanding the possible motivations of these key actors, we hoped to better understand the key factors in environmental governance, such as political arrangements and incentives.

## 2. Literature Review and the Theoretical Framework: Understanding Carbon Emissions in the Context of the Chinese Political Environment

### 2.1. Carbon Emissions Trading System

Traditional regulation tools and policies such as environmental tax and industrial energy prices were inefficient due to information asymmetry [12]. Existing studies have discussed that using a market-oriented mechanism is more efficient and economical on the realization of carbon emissions reduction compared to policies directly led by government, such as environmental protection administrative penalties and pollution charges [13]. Therefore, it is imperative to design effective market-oriented regulation policies for reducing carbon emissions.

During the past decades, carbon emissions trading systems have become important tools in market-oriented environmental regulation to address the issues of inefficient carbon emission allocation. Building on Coase's option theory, John Dales proposed a system of carbon emissions trading in 1968 [14]. Introducing property rights into environmental pollution control was meant to internalize the cost of carbon dioxide emissions [15]. The

literature on carbon trading mechanisms has mainly focused on the effects of regional carbon emissions reduction programs in developed countries. The markets were often the research objects, especially carbon trading in Europe's EU-ETS program, which operates through auctions. To better quantify and empirically study the utility of carbon trading mechanisms, scholars began to use different models such as the dynamic decision-making model, the general equilibrium model, and the network analysis model [16]. For example, Dong et al.'s study of China's carbon emissions trading policy found that market scale and reduction costs were negatively correlated [17]. Martin et al. focused on the impact of emissions trading on enterprise [18]. Other scholars used different models to explore the impacts of carbon trading on other affected groups.

The effectiveness of carbon emissions trading is relatively unexamined. The majority of the existing studies are qualitative research, predictive simulations, or are focused on specific industries or regions. At the national level, empirical research is scarce. Few empirical studies have analyzed the effectiveness of pilot emission trading systems on politics. Additionally, the performance of carbon emissions trading systems remains controversial. Even so, scholars have generally determined that China's carbon emissions trading system has successfully promoted carbon emission reductions [19].

Despite the fact that these studies investigated the efficiency of policies, the results they reached differed substantially due to discrepancies in their data and techniques. The emission reduction results of these systems under diverse viewpoints varied since some researchers concentrated on various core objects. Others adopted different policy evaluation methods, making it difficult to obtain unbiased estimates for the variable of carbon emissions. Similarly, the literature also provides a theoretical framework for the study of carbon emissions trading systems. Theoretical arguments for the emission reduction effects of these systems can certainly be made. However, they are frequently biased and ineffective. Stated another way, the majority of studies have focused on the direct impact of carbon emission trading systems while disregarding politics.

### 2.2. Political Mobility in the Context of China's Politics

Government officials actively participate in environmental governance as managers of society and advocates for government policies. The reasons behind their acts will inevitably have an impact on both their work performance and the government's. In most Western democracies, the political reputation model can explain how political incentives affect environmental governance [20]. Officials may decide to modify tax and environmental governance policies in accordance with the preferences of their constituencies to protect their chances of being re-appointed or re-elected.

Different from Western democracies, local officials in China are appointed by higher-level authorities, but local governments nevertheless retain a lot of power. Sometimes referred to as federalism with Chinese characteristics, this division of authority makes China's economic growth possible through administrative and fiscal decentralization [21]. However, this theory relies upon a high degree of institutional stability, which creates its own motivating effect. The Chinese promotion tournament theory proposes a Principal-agent relationship among the levels of government. China's administrative level-by-level contract system represents a typical type of strong incentive contract from the perspective of economics [22]. The central government, as the employer, holds political tournaments among provincial governments by virtue of personnel appointment or the recommendation power of the administrative head. From this point of view, in a political environment where vertical contracting and horizontal competition are highly unified, the extensiveness and unity of local government power provided by the administrative subject contract system gives the administrative subject enough space to play a significant role. Many studies have found that economic growth is the primary indicator of political promotion. Hence, local governments frequently devote their main resources to fostering economic growth [23]. Projects that result in short-term economic growth will be approved by local officials, regardless of any long-term environmental implications.

This situation is changing. Environmental protection has been a focal point since the Chinese government shifted its goals. Environmental protection is now included in the performance evaluation of local officials. This is important because the promotion tournament incentive structure has had severe consequences for environmental governance, such as global warming [24], excessive energy consumption and environmental pollution. Therefore, understanding the initial impact of carbon emissions on local officials is important for China's policy response to climate change. Strangely, the role of local officials in emissions reduction has largely been ignored by researchers.

*2.3. Theoretical Framework and Research Hypothesis*

2.3.1. The Performance of the Carbon Emissions Trading System

The implementation of a carbon emissions trading policy may improve the environmental circumstances in the pilot region in a market-oriented way. Because carbon emissions can be measured, it is possible to compare regional carbon emission controls on a horizontal scale. While pursuing political promotion, local officials in areas with fewer emission reductions experience higher pressure. Simply said, regional environmental quality improvements can help officials get promoted. Therefore, local officials will compete with other officials at the same level on environmental protection issues in order to gain promotion opportunities [25]. Where once they competed to demonstrate economic progress, officials must now show environmental stewardship. At the same time, poor performance in an environmental quality evaluation will likely attract the attention of environmental protection departments.

From the industry perspective, carbon emissions trading policy has improved the incentives to reduce production costs through emission reduction. Enterprises have also gained a measure of autonomy through trading carbon emissions. There will be significant external pressure on the government in regions with high carbon emissions to improve local environmental governance. Under the joint action of the above two subjects, local governments will make every effort to implement the carbon emission trading policy, improve regional environmental quality, strengthen regional environmental governance and supervision, and strive to reverse the bad environmental situation in the region. This behavior is conducive to promoting the reduction of carbon dioxide emissions. Based on this, we propose the first hypothesis:

**Hypothesis 1.** *China's carbon emissions trading system has a significantly positive effect on reducing China's carbon emissions.*

2.3.2. Effect of Political Mobility on Carbon Emission Reduction Performance

With the opening up of China in the late 1970s, local officials have taken an active role in constructing a new economic system, developing the private economy and reforming the local government performance evaluation system. For the carbon emission reduction scheme, efficiency incentives were to be provided to local officials. It was necessary to link reductions to political promotion. Previous studies of provincial officials noted that the performance of local officials may be affected by their tenure, which is highly influenced by economic performance. Chen et al. focused on the relationship between performance evaluation and the tenure of local officials [26]; their tenure depended on local economic performance. Similarly, He et al. proved that the higher the completion ratio of economic growth targets, the higher the probability of promotion, and the higher the probability of political promotion in the case of higher economic growth targets [27].

Moreover, according to the multi-task principal-agent theory, when there are multiple tasks to accomplish local officials always decide on the action with the highest expected utility and obvious results. The period for achieving political goals through economic growth is brief and simple to track. Environmental protection requires a long period and high investment. Hence, rational local government officials will prioritize economic growth and reduce investment in environmental governance due to the constraints of multi-

dimensional political goals assessment and limited financial resources. This will in turn affect environmental quality. Thus, due to the externality characteristics of environmental protection, local officials may choose to disregard environmental issues. Extensive growth patterns even have a crowding out effect on environmental governance expenditure, which is not conducive to the improvement of environmental quality. The central government could change this. Local officials might focus on environmental regulation performance if it had an impact on their political tenure [28]. This leads to our second set of hypotheses:

**Hypothesis 2-a.** *Carbon emission reduction performance is negatively associated with the tenure of local officials.*

**Hypothesis 2-b.** *Carbon emission reduction performance is positively associated with the tenure of local officials.*

How do local officials get into positions of power in China? Previous studies provided two models to explain political mobility: the factional model and the performance model. The factional model considers personal relationships to central leaders as the key to career success [29]. However, just as Teiwes advocated the utility of the factional concept, the efficiency of the factional model has not been systematically tested [30]. The performance model is the current mainstream explanation. Bo insisted on using the performance model to challenge the traditional factional model [31]. Based on the performance model, the political mobility of local officials mainly included the following five movements: promotion (vertical mobility from a lower rank to a higher one); demotion (includes demotions and purges or dismissals); lateral transfer (horizontal mobility without change in rank); retirement; and continuing without movement. Local officials faced different promotion incentives depending on their experiences, which might affect their environmental governance performance [32]. In this study, we sought to observe whether and how political mobility incentives affect carbon emission reduction performance, so we chose promotion and lateral transfer as the main types for our research.

Officials who are promoted within their home region are more familiar with local circumstances and therefore tend to set higher performance targets. Compared to locally promoted officials, officials transferred from higher authorities and other localities tend to avoid more challenging tasks or taking risks to achieve higher goals due to their unfamiliarity with the area [33]. Therefore, we assumed that locally promoted officials were more likely to set higher government performance goals than transferred officials.

In China's public personnel system, if a central government official was appointed to be a governor, he/she would probably regard this experience as compulsory training that would enable further promotion within the bureaucratic system. Thus, local officials connected with the central government have more chances of promotion. As a result, officials with close ties to the central government tend to make more conservative decisions to safeguard their chances of promotion [34].

The paths taken by officials transferred horizontally from other provinces are more complicated. Some scholars believe that the political mobility system helps higher-level government officials accomplish their long-term strategic goals. The active mobility of officials avoids nepotism, which is typically caused by the long-term employment of officials in one place. In addition, officials also tend to set higher goals as they gain more ability by learning, exchanging experiences and updating their governance concepts. Therefore, they usually formulate strategies to show their unique views, highlight their potential, and help realize the intention of coordinating regional development. This is reflected in the completely different emission reduction governance methods of those who were locally promoted and those who were transferred from the central government. Therefore, contrary to previous conclusions, there is evidence that officials transferred to new provinces can achieve higher performance [35].

Hence, we set dummy variables and propose a third set of hypotheses by taking "officials promoted from other provinces" as the reference group:

**Hypothesis 3-a.** *Compared with officials in the control group, locally promoted officials will have higher carbon emission reduction performance.*

**Hypothesis 3-b.** *Compared with officials in the control group, officials transferred from the central government will have lower carbon emission reduction performance.*

**Hypothesis 3-c.** *Compared with officials in the control group, officials transferred from other provinces will have higher carbon emission reduction performance.*

Additionally, educational background is important for cadre selection and political mobility in China, and local officials are therefore often keen to obtain a high academic level [36]. Thus, an improvement of the educational background of local officials results in a higher likelihood of cadres being more aware of the basic needs of citizens, understanding the connotations of economic development more openly and comprehensively, and paying more attention to the coordinated development of the environment and the economy. Based on this, we proposed the following hypothesis:

**Hypothesis 4.** *Local officials with a higher level of education will show more resolve to achieve carbon emission reductions.*

The above theoretical hypotheses were proposed through theoretical analysis. The theoretical research framework of this study was obtained and tested using the empirical analysis described below.

## 3. Materials and Methods

### 3.1. Statistical Modeling

To assess the net effect of the carbon emissions trading policy, we measured the difference between the state of the pilot provinces following intervention by the carbon emissions trading mechanism and the assumed state without policy trials. The latter kind of state, known as the counterfactual state, is not observable, but can be estimated via comparison with a control group (i.e., the non-pilot provinces) [37].

The carbon trading mechanism trials in the pilot provinces commenced in 2013, so this paper takes 2013 as the policy implementation year. From 2013 to the national introduction of the carbon market in 2017, the policy only applied to the pilot provinces; all non-pilot provinces were unaffected. Thus, the non-pilot provinces and cities were taken as the control group. Based on Hypothesis 1, we used the DID method to compare the carbon reduction performance before and after the launch of the carbon trading policy. Since the study has two groups with divided research objects, it can be considered a "quasi-experimental" design. The DID method is suitable for the causal effect estimation of the quasi-experiment because it can avoid endogeneity. That is, it can effectively control the interaction effect between the explained variable and the explanatory variable. In the DID model of panel data, the exogenous explanatory variable can be used to control the unobservable individual heterogeneity between samples. It can also control the influence of unobservable factors which change with time, so it can produce an unbiased estimation of the policy effect. To ensure the robustness of the results, we verified the estimation results through different test methods.

Based on the DID model expressed in econometrics, we established the following model Formula (1) to evaluate the emission reduction performance of the carbon emissions trading policy:

$$Y = \alpha + \beta_1 \times policy \times (G_i \times D_t) + \Sigma\beta_j \times control + \gamma G_i + \lambda D_t + \varepsilon_{it} \tag{1}$$

$Y$ is the explained variable carbon emission and Control represents a series of control variables. Based on the above theoretical assumptions, the control variables included in the policy evaluation Model (1) are the economic, social and political factors that affect carbon emissions. $G_i$ is the grouping dummy variable. If $G_i = 1$, it is a pilot province, which is the intervention group; if $G_i = 0$, it is a non-pilot province, which is the control group. This parameter indicates that even without the influence of this policy, there would still be unchangeable differences between the two comparison groups due to some other uncontrollable factors. $D_t$ is the staging dummy variable (after policy implementation $D_t = 1$, before policy implementation $D_t = 0$), representing the time difference before and after policy implementation. The interaction term $G_i \cdot D_t$ (policy) represents the net effect of the carbon emissions trading policy. $\varepsilon_{it}$ is the random interference term. If the coefficient $\beta_1$ of the policy is not significantly equal to 0, it means that China's carbon emissions trading policy has a significant impact on carbon emission reduction. However, if $\beta_1 = 0$, this indicates that China's carbon emissions trading policy is invalid.

Based on the analysis and assumptions of the relevant variables affecting carbon emissions, we constructed multiple regression models to estimate the coefficients. Model (2) is set as follows:

$$Y = \alpha + \beta_1 policy + \beta_2 pergdp + \beta_3 population + \beta_4 energy + \beta_5 secindustry + \beta_6 tenure + \beta_7 edu \\ + \beta_8 local + \beta_9 center + \beta_{10} intertrans + \varepsilon_{it} \tag{2}$$

$Y$ is the explained variable carbon emission, $\alpha$ is a constant term, $\beta_i$ ($i = 1, 2, 3, \ldots, 10$) represents the regression coefficient of the explanatory variable and the control variable and $\varepsilon_{it}$ is the random error term. Specifically, *policy* represents the implementation of a carbon trading policy, while *tenure*, *edu*, *local*, *center* and *intertrans* are the variables reflecting officials' characteristics. For *pergdp*, *population*, *energy* and *secindustry* denote the control variables. Table 1 provides the detailed variable definitions.

**Table 1.** Definitions of variables of used to measure the effectiveness of China's carbon emissions trading system and their measurement methods.

| | Variable Name | Variable Meaning | Variable Operation | Data Source | Direction |
|---|---|---|---|---|---|
| Dependent variable | Emission | Carbon emission | Unit: million tons | CEADs | / |
| Political factors | Tenure | Official's tenure | Measured by number of years that the official holds his/her position | China Political Elite Database (CPED) | / |
| | Edu | Official's education background | Measured using sequencing variables, divided into 1 (senior high school or below), 2 (junior college or bachelor's degree), 3 (master's degree), 4 (doctorate) | China Political Elite Database (CPED) | − |
| | Local | Promoted form local province | Vertical promotion from a local province is recorded as "1", otherwise it is recorded as "0". | China Political Elite Database (CPED) | − |
| | Center | Transferred from central government | Horizontal transfer from central government is recorded as "1", otherwise it is recorded as "0". | China Political Elite Database (CPED) | + |
| | Intertrans | Transferred from another province | Horizontal transfer from another province is recorded as "1", otherwise it is recorded as "0". | China Political Elite Database (CPED) | |

**Table 1.** *Cont.*

| | Variable Name | Variable Meaning | Variable Operation | Data Source | Direction |
|---|---|---|---|---|---|
| Economic and social factors | Pergdp | GDP per capita | Unit: CNY/person | National Bureau of Statistics | + |
| | Population | Resident population at year end | Unit: ten thousand | National Bureau of Statistics | + |
| | Energy | Energy consumption per unit of GDP | Ratio of regional energy consumption to regional GDP (unit: ton of standard coal/10,000 CNY) | China Statistical Yearbook | + |
| | Secindustry | Secondary industry added value | Ratio of added value of secondary industry to regional GDP | National Bureau of Statistics | + |

Note: The + and − in the last column indicate that the expected relationship between independent variables and dependent variables is positive or negative, respectively.

*3.2. Data and Variables*

Considering the comparability and accessibility of the known data, we selected the carbon emissions of 30 provincial administrative units in China as the explanatory variables, including Shenzhen, Beijing, Tianjin, Shanghai, Chongqing, Guangdong and Hubei, all of which have implemented the pilot carbon trading system. In 2003, China proposed the concept of "scientific development". Given the nation's authoritarian political environment, this concept was set to become the new direction of local governments. As a result, local officials attached great importance to the concept of environment protection at this time. In the future, 2003 may be seen as an important turning point in the trajectory of carbon emissions. Given this history, we examined the changes in carbon emissions in selected provinces from 2004 to 2015.

Since the calculation method for carbon emissions is complex and involves a highly specific discipline, the carbon emissions data for each province in this study were obtained from the database website China Emission Accounts and Datasets (CEADs). The independent variable and control variable data were obtained from the following databases: the Chinese Statistical Yearbook, the National Bureau of Statistics, and the Chinese Political Elite Database.

The variables of carbon emissions, economic factors, and social factors were continuous variables obtained from the databases mentioned above, whereas the variables of political factors were obtained by the author through the selected database and quantified for measurement. Among them, the length of an official's term was defined using the method in the existing relevant literature (i.e., the number of years from the beginning of the post to final departure from the position). Since the official appointment and departure time usually occurs in a certain month of a certain year, if the official takes office in the first half of the year (January–June), the year was taken as the starting year of his/her appointment; otherwise, the official term was calculated from the next year. Based on the previous studies, officials' education was measured by sequencing variables and we divided them into 1 (senior high school or below), 2 (junior college or bachelor's degree), 3 (master's degree) and 4 (doctorate) [38]. The sources of the provincial governors were measured by setting dummy variables, in which the reference group was composed of officials promoted from other provinces.

Previous studies have shown that carbon dioxide emissions are affected by various economic and social factors. The consensus is that there is a negative correlation between economic and social development and carbon emissions. This study considers influencing factors as control variables, including the variables of per capita GDP, population, GDP energy consumption and added value of secondary industries.

Per capita GDP (per GDP) was used to reflect the level of economic development in a region. As for the correlation between the economy and carbon dioxide emissions, the mainstream view supports an inverted "U" relationship, namely, the Environmental Kuznets Curve [39]. This means that the environmental quality degrades with an increase in income within a certain range, then improves after the income reaches a certain level. However, another stream of research opposes these findings and rejects the Environmental Kuznets Curve [40]. No consensus has been reached regarding the relationship between economic growth and an increase in carbon dioxide emissions, which may be due to the different economic development stages of the research objects. Because the levels of economic development differed from region to region within China, we believe that provinces and cities with a higher per capita GDP have a greater demand for economic development, and rapid economic development will increase carbon emissions; thus, per capita GDP is positively correlated with greenhouse gas emissions.

Regarding population, Birdsall stated that population growth can affect greenhouse gas emissions in two ways [41]. First, a larger population will have a higher energy demand, which is accompanied by an increase in carbon emissions. Second, rapid population growth often leads to environmental destruction, which is not conducive to the reduction of carbon dioxide. Kaya established the correlation between greenhouse gas emissions and population through the identity of factor decomposition [42]. Through the modified Kaya identity, STIRPAT, and other models, follow-up research has shown that there is a stable and long-term positive correlation between population growth and the urbanization process and carbon emissions. Therefore, we expect a positive correlation between population and carbon emissions.

In terms of GDP energy consumption, the production of carbon dioxide mainly comes from energy consumption, and the carbon emissions of a region are inevitably affected by local energy utilization. The energy consumption per unit of GDP measures the energy utilization of provinces and cities. The measurement unit is the energy consumption per ten thousand CNY of GDP, which reflects the economic benefits of energy consumption. Based on this, we expected a positive correlation between GDP energy consumption and carbon emissions.

As for the added value of secondary industries, empirical studies have proven that secondary industries play an important role in carbon emissions, and the adjustment of industrial structure is an important driving factor of changes in carbon emissions [43]. We used the secondary industry value added ratio to measure the level of industrial structure. Compared with primary and tertiary industries, the energy demand of secondary industries is relatively high. Therefore, it was more intuitive to select the added value of the secondary industries. Based on this, we expected the value-added ratio of the secondary industries to be proportional to carbon emissions. The variables are shown in Table 1 above.

### 3.3. Descriptive Statistics

Table 2 shows the descriptive statistical analysis of each variable conducted in this study. The diagnosis results of collinearity among variables shows that the variance expansion factor (VIF) is much smaller than 10, indicating that there was no significant collinearity between variables. Based on this, further conclusions were obtained using the econometric regression model.

**Table 2.** Descriptive statistics of main variables.

| Variable Name | Observations | Mean | Standard Deviation | Minimum Value | Maximum Value |
|---|---|---|---|---|---|
| Tenure | 360 | 3.186 | 1.854 | 1 | 10 |
| Edu | 360 | 2.743 | 0.637 | 2 | 4 |
| Local | 360 | 0.700 | 0.459 | 0 | 1 |

**Table 2.** *Cont.*

| Variable Name | Observations | Mean | Standard Deviation | Minimum Value | Maximum Value |
|---|---|---|---|---|---|
| Center | 360 | 0.156 | 0.362 | 0 | 1 |
| Intertrans | 360 | 0.097 | 0.297 | 0 | 1 |
| Emission | 360 | 288.91 | 234.56 | 5.80 | 1553.80 |
| Pergdp | 360 | 33,215.43 | 21,617.04 | 4317 | 107,960 |
| Population | 360 | 4413.62 | 2651.96 | 539 | 10,849 |
| Energy | 360 | 1.166 | 0.651 | 0.298 | 4.324 |
| Secindustry | 360 | 0.471 | 0.077 | 0.197 | 0.590 |

## 4. Results and Discussions

### 4.1. Analysis of Parallel Trend Test and DID Results

In this study, based on Formula (1) we used the DID model to test the impact of the carbon emissions trading policy on carbon emissions. However, to be consistent, a key assumption for the DID estimator was the parallel trend hypothesis, also known as the common trend assumption. It states that if there is no policy impact, then the trend of the treatment and control should be parallel and have the same time trend. Otherwise, if the model fails to meet this assumption, this estimation will be biased. Therefore, a parallel trend check was performed for the DID model.

As shown in Figure 1, we drew a comparison diagram of changes in the carbon emission trend between pilot provinces and non-pilot provinces to illustrate the changes before and after the pilot trading system. Figure 1 intuitively shows that prior to the implementation of the carbon emissions trading system in 2013, the growth trend of carbon emissions in the two groups was almost identical; there was no systematic difference over time. However, after initiation of the pilot in 2013, the trends in the pilot provinces changed and carbon emissions were stable or even declining. The non-pilot provinces maintained their pre-2013 growth trends. Therefore, we believe that these data meet the premise of using the DID method, which can successfully identify the net effect of a carbon emissions trading mechanism.

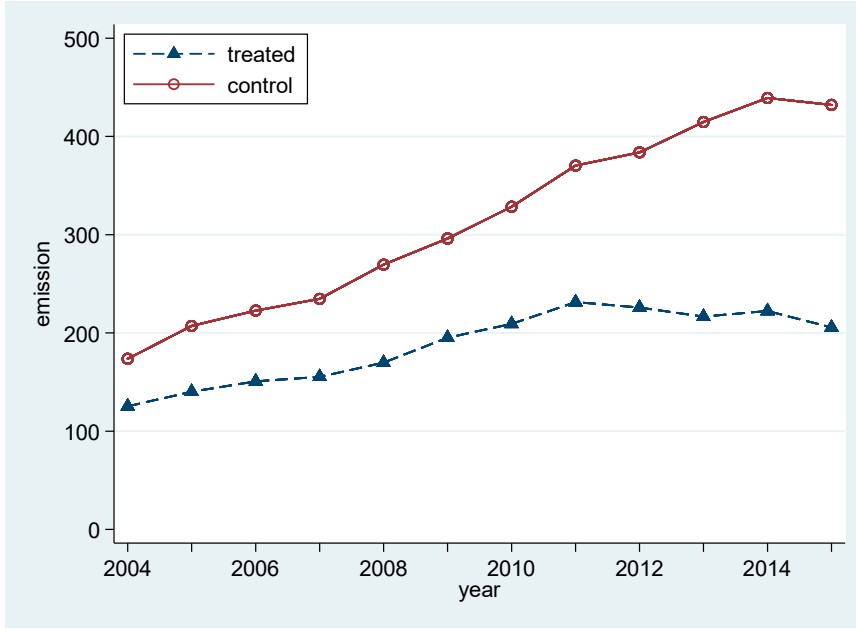

**Figure 1.** Time trend of carbon emissions in pilot and non-pilot provinces from 2004 to 2015.

The carbon emissions trading system pilot provides us with a quasi-natural experiment. In this study, the effect of this policy was analyzed using the DID method, and the results are shown in Table 3.

**Table 3.** Effect test of the carbon trading mechanism pilot.

| Variable | Model 1 | Model 2 |
| --- | --- | --- |
| Policy | −109.12 ** (41.93) | −147.50 ** (54.24) |
| Control variables | No | Yes |
| Individual effect | Yes | Yes |
| Time effect | Yes | Yes |
| Constant | 262.79 *** (36.62) | −519.70 (262.49) |
| N | 360 | 355 |
| $R^2$ | 0.126 | 0.535 |

Note: *** $p < 0.01$, ** $p < 0.05$; Outside the brackets are the coefficients, and within the brackets are the robust standard errors aggregated at the provincial level.

In Table 3, Model 1 is the estimated result without the control variables, while Model 2 is the estimated result by adding in the control variables. The regression coefficient of the carbon emissions trading system policy is significantly negative at the 5% level, indicating that the carbon trading system has a significant effect on carbon emissions reduction in the pilot provinces. Compared with the non-pilot provinces, the carbon trading system reduced carbon emissions in the pilot provinces by an average of more than 140 million tons. Hypothesis 1 is therefore verified, which indicates that China's carbon emissions trading system has a significantly positive effect on reducing China's carbon emissions.

### 4.2. Effect Test of Related Variables

#### 4.2.1. Regression Results

Based on Formula (2), the regression results are shown in Table 4. Model 3 is a regression result that includes only economic and social factors; Model 4 presents the regression result that includes only the factors of political mobility; finally, Model 5 is the whole model with both dependent and control variables.

**Table 4.** Influence of the carbon emission trading policy on carbon emission reduction performance.

| Variable | Model 3 | Model 4 | Model 5 |
| --- | --- | --- | --- |
| $G_i \cdot D_t$ | −155.61 *** (43.20) | −100.47 (78.39) | −130.95 ** (48.86) |
| Tenure | | 4.28 (9.21) | −6.90 (7.45) |
| Edu | | 16.76 (40.98) | −1.10 (35.38) |
| Local | | −79.93 (76.27) | −29.79 (56.01) |
| Center | | −9.15 (94.14) | −6.68 (61.48) |
| Intertrans | | −209.74 ** (85.33) | −152.06 * (72.94) |
| Pergdp | 0.005 *** (0.001) | | 0.005 *** (0.001) |
| Population | 0.045 *** (0.010) | | 0.044 *** (0.011) |
| Energy | 79.55 * (42.94) | | 88.53 ** (39.20) |
| Secindustry | 651.99 *** (226.94) | | 776.21 *** (224.34) |
| Constant | −457.54 *** (144.57) | 365.58 (264.82) | −509.95 ** (267.16) |
| N | 360 | 355 | 355 |
| $R^2$ | 0.491 | 0.166 | 0.527 |

Note: *** $p < 0.01$, ** $p < 0.05$, * $p < 0.1$; The robust standard errors aggregated at the provincial level are reported in the parentheses.

The results show that local official tenure has no significant relationship with carbon emissions reduction. As a consequence, Hypothesis 2 cannot be proved. The results also indicate that only the officials transferred from other provinces had a significant negative impact on carbon emissions, thus supporting Hypothesis 3-c. Officials who were promoted from within their provinces tended to have higher carbon emissions reduction performance

than those from the central agencies; this is consistent with the hypothetical direction, but the differences are not significant. Thus, the effectiveness of carbon emissions for officials promoted from different sources suggests that political mobility influences the provincial government leaders' environmental governance performance. Although the influence coefficient of education in Model 5 on the performance of carbon emissions is negative, it is not significant so Hypothesis 4 cannot be proved. Generally, these results validate the logic behind the actions of government officials. However, most of them failed to pass the significance test because local officials usually must pursue many vague and pluralistic goals. Environmental performance is not yet the core focus of China's official assessment indicators. As such, the exact role that an official's characteristics play in environment performance cannot be well understood.

The results of the control variables related to the economy and society were all in line with expectations. The coefficient of the per capita GDP of the region was significantly positive at the 1% level. This indicates that China's economic development level is still in a relatively early stage, and is highly dependent on the sacrifice of environmental resources. China's economic growth has not passed the stage of "exchanging environmental quality for economic growth". The coefficient of the variable population is significantly positive, which indicates that China, as a country with a large population, undoubtedly contributes significantly to the emission of carbon dioxide. The coefficients of energy per unit of GDP and value added of secondary industries are also significantly positive, which is also in line with general experience. Currently, China's energy structure is not especially advanced, and the further development of secondary industries will still primarily depend on energy consumption. In sum, the conflict between economic development and the capability of environment governance has become increasingly evident, which has caused a negative impact on the overall level of carbon emissions.

### 4.2.2. Interaction Effect Test

To test the interaction effect between the political factors and carbon emissions, this study examined the moderating effects of official sources on the correlation between tenure and career path on carbon emissions by referring to the test method of interaction effects in the regression analysis. Table 5 reports the test results.

**Table 5.** Regression results of interaction effects of the variables regarding political mobility.

| Variable | Model 6 | Model 7 | Model 8 |
|---|---|---|---|
| Local | 28.59 (28.12) | | |
| Local $\times$ tenure | 26.80 ** (12.10) | | |
| Local $\times$ tenure$^2$ | −0.12 (1.67) | | |
| Center | | 52.70 (39.73) | |
| Center $\times$ tenure | | −20.54 * (11.78) | |
| Center $\times$ tenure$^2$ | | −10.13 * (5.07) | |
| Intertrans | | | −120.27 ** (44.67) |
| Intertrans $\times$ tenure | | | −22.79 ** (9.72) |
| Intertrans $\times$ tenure$^2$ | | | −2.52 (8.03) |
| Constant | −516.80 ** (188.01) | −451.45 ** (165.55) | −467.97 *** (155.17) |
| N | 355 | 355 | 355 |
| R$^2$ | 0.502 | 0.496 | 0.512 |

Note: *** $p < 0.01$, ** $p < 0.05$, * $p < 0.1$; Outside the brackets are the coefficients, and within the brackets are the robust standard errors aggregated at the provincial level.

The regression results show that the interaction effect between the source and the term of the selected governor has passed the significance test, thus verifying the regulatory effect of official sources on government performance. The interaction effect between local promotion and tenure was significantly positive, while the transfers of central and provincial and tenure were both significantly negative. That is to say, the longer the tenure of the local official who was promoted, the more carbon emissions tended to increase. On

the other hand, officials moving from the central government and those moving to a new province had the opposite result. This indicates that government officials from different sources have different carbon emissions reduction performance levels.

### 4.3. Robustness Test

In this study, robustness tests were carried out, including the widely used single-difference test, counter-fact test and matching method, to ensure the reliability of the results.

### 4.3.1. Single Difference Test

According to the traditional treatment method, we used the single difference method to estimate the impact of the carbon emissions trading system on carbon emissions. The regression results are shown in Table 6. As expected, the single difference test overestimates the effectiveness of the carbon emissions trading system. Meanwhile, it proves the validity of our DID estimation results.

**Table 6.** Single difference test estimation results of the impact of the carbon emissions trading system.

| Variable | Model 9 | Model 10 |
|---|---|---|
| Policy | −127.08 * (62.92) | −146.87 ** (62.75) |
| Control variables | No | Yes |
| Individual effect | Yes | Yes |
| Time effect | No | No |
| Constant | 314.32 *** (43.54) | 49.14 (25,765.05) |
| N | 360 | 343 |
| $R^2$ | 0.047 | 0.287 |

Note: *** $p < 0.01$, ** $p < 0.05$, * $p < 0.1$; Outside the brackets are the coefficients, and within the brackets are the robust standard errors aggregated at the provincial level.

### 4.3.2. Counterfactual Test

Drawing on the robustness of the test methods used in previous research, we conducted counterfactual tests by changing the setting of policy pilot provinces. If the coefficient of the carbon emissions trading system was still significantly negative, it indicated that carbon emission decline may be a result of other policies or random factors, but not necessarily a result of the carbon emissions trading mechanism. The test results are listed in Table 7. Six pilot provinces were chosen: Zhejiang, Anhui, Jiangxi, Shandong, Henan and Hunan. Model 12 and Model 11 are the estimated results with and without the addition of control variables, respectively. Six additional pilot provinces were then chosen: Yunnan, Qinghai, Shaanxi, Gansu, Ningxia and Xinjiang. Model 14 and Model 13 are the estimated results with and without the addition of control variables, respectively. The coefficients of the four models were not significant. This indicates that the downward trend of carbon emissions in the pilot provinces and selected cities was caused by the carbon emissions trading system rather than other factors.

**Table 7.** Results of the counterfactual test.

| Variable | Model 11 | Model 12 | Model 13 | Model 14 |
|---|---|---|---|---|
| Policy | −21.39 (45.01) | −79.05 (52.50) | −5.91 (58.28) | 34.99 (61.23) |
| Control variables | No | Yes | No | Yes |
| Individual effect | Yes | Yes | Yes | Yes |
| Time effect | Yes | Yes | Yes | Yes |
| Constant | 221.93 *** (28.81) | −571.20 * (264.69) | 270.84 *** (36.23) | −543.08 ** (307.57) |
| N | 360 | 355 | 360 | 355 |
| $R^2$ | 0.104 | 0.495 | 0.123 | 0.502 |

Note: *** $p < 0.01$, ** $p < 0.05$, * $p < 0.1$; Outside the brackets are the coefficients, and within the brackets are the robust standard errors aggregated at the provincial level.

4.3.3. Matching Test

Although the carbon emissions trading system pilot successfully met the conditions necessary to be considered a quasi-experiment, the choice of pilot provinces may have had endogenous problems, leading to selection bias. Matching pre-processing prior to estimating the causal effect can solve this problem under non-random experimental conditions and thus overcome the problem of choice. The idea of the matching method was derived from the matching estimator of the counterfactual framework. The basic idea was to find the individual belonging to the control group so that the value of the measurable variable was as close as possible to the value of the experimental group. That is, the matching method could be used to separate a matching sample with relatively balanced covariates to find randomized experimental samples hidden in the observed data. Therefore, it became obvious which samples could be used to estimate the matching estimators, and the resulting matching estimators could be interpreted as the individual causal effects [44]. In addition, the use of matching samples for analysis can make the estimation results more robust and less sensitive to functional forms, as well as reduce the model dependence of the subsequent regression analysis to estimate causal effects.

This study used the ebalance command to perform matching, and the matching principle was the entropy balancing method. The entropy balancing method is a multivariate weighting method which overcomes some of the problems encountered in traditional matching methods, such as nearest neighbor matching and propensity score matching. Traditional matching methods involve a long matching process in which it is difficult to balance all the covariates. In practice, the matching covariate balance is often low.

The entropy balancing method directly integrated the covariate balance into the weight function used to adjust the control group data. Based on the maximum entropy weight, the data can satisfy as large a balance constraint as possible. Compared with the traditional matching method, the entropy balance method has some advantages. Because the weight value in the entropy balance is directly adjusted to the known sample moment, the covariate balance of the traditional processing method under the constraint of a given moment is improved; therefore, a balance test is not required. In addition, because the entropy balance weight changes smoothly between each element, this method retains more information than other methods and the estimated result after matching is more accurate.

Table 8 shows the balanced results of the matched variables. As mentioned above, due to the particularity of the value of the political and business relationship in the variable, multiple matching methods still cannot achieve effective matching, which increases the overall matching difficulty. Therefore, this variable is not included in the matching process. Compared with the results before matching, the deviations of the matched treated group and the control group are effectively reduced, and the mean and standard deviation of the matching variables are also very close, indicating that the selected matching method is appropriate. Therefore, the estimation results are reliable.

**Table 8.** Results of covariate equilibrium.

| Variable | Means | | | Variances | | | Skewness | | |
|---|---|---|---|---|---|---|---|---|---|
| | Treated | Control | | Treated | Control | | Treated | Control | |
| | | Pre | Post | | Pre | Post | | Pre | Post |
| Tenure | 3.972 | 3.011 | 3.972 | 4.816 | 2.933 | 3.407 | 0.696 | 0.828 | 0.698 |
| Edu | 2.917 | 2.700 | 2.917 | 0.416 | 0.395 | 0.084 | 0.075 | 0.330 | −2.566 |
| Local | 0.639 | 0.717 | 0.639 | 0.234 | 0.204 | 0.232 | −0.578 | −0.965 | −0.578 |
| Center | 0.056 | 0.184 | 0.056 | 0.053 | 0.151 | 0.053 | 3.881 | 1.663 | 3.880 |
| Intertrans | 0.181 | 0.071 | 0.180 | 0.150 | 0.066 | 0.149 | 1.661 | 3.351 | 1.661 |
| Pergdp | 52,746 | 28,448 | 52,744 | $7.67 \times 10^8$ | $2.77 \times 10^8$ | $3.77 \times 10^8$ | 0.338 | 1.020 | 0.179 |
| Population | 4012 | 4467 | 4012 | 9,668,793 | 6,210,363 | 8,951,918 | 1.123 | 0.397 | 0.434 |
| Energy | 0.771 | 1.266 | 0.771 | 0.092 | 0.467 | 0.160 | 0.885 | 1.521 | 1.432 |
| Secindustry | 0.438 | 0.480 | 0.438 | 0.010 | 0.005 | 0.013 | −1.123 | −1.235 | −0.872 |

After matching, we again estimated the effectiveness of the carbon emissions trading system on emissions reduction. The results are shown in Table 9. Model 15 is the estimated result without the addition of control variables after matching, and Model 16 is the estimated result after considering other relevant influencing factors after matching. Regardless of whether control variables were added, the coefficient of policy was significantly negative. That is, the carbon emissions trading system promoted carbon emission reduction in pilot provinces, which was consistent with the estimation result of DID. In other words, after considering the problem of sample selection bias, Hypothesis 1 of this paper was still valid, which further indicated that the DID estimation results were robust and reliable.

**Table 9.** Net effect test of carbon trading system after matching.

| Variable | Model 15 | Model 16 |
|---|---|---|
| Policy | −173.97 ** (68.45) | −128.21 *** (25.06) |
| Control variables | No | Yes |
| Individual effect | Yes | Yes |
| Time effect | Yes | Yes |
| Constant | 361.22 *** (66.98) | 224.44 (150.61) |
| N | 355 | 355 |
| $R^2$ | 0.140 | 0.790 |

Note: *** $p < 0.01$, ** $p < 0.05$; Outside the brackets are the coefficients, and within the brackets are the robust standard errors aggregated at the provincial level.

## 5. Conclusions and Policy Implications

China attaches great importance to and plays a constructive role in climate change. In the Paris Agreement, the Chinese government committed to the target of achieving peak carbon emissions by 2030 [45]. The central government has gradually improved the mechanisms of carbon emissions reduction by implementing a market-oriented carbon trading system. The environmental regulation tools and the concept of low carbon have made progress in controlling carbon emissions and adapting to climate change.

Starting with the development of carbon neutrality strategies in China, this paper constructed a quasi-natural experiment based on the DID method to evaluate the effectiveness of carbon emission reductions by China's carbon trading system from a perspective of political mobility with panel data for 30 provinces in China from 2004 to 2015, and it conducted a robustness test using the counterfactual test and matching test to further test the robustness of the results. This paper mainly makes contributions in three aspects. First, on the theoretical side, in addition to examining the impact of the carbon trading system on carbon emissions, this study's findings help to explain the impact of political factors on environmental governance. Second, on the methodology side, the DID model and various robustness tests this paper conducted have expanded the processes available to evaluate the policy's net effect on carbon emissions and revealed the effect of officials' incentives to reduce emissions. Finally, on the practical side, this paper helps the central government to gain a deeper understanding of carbon emissions trading, so as to design more effective and targeted strategies for motivating officials and supervising policy implementation to promote carbon neutralization and environmental development. This paper aimed to explore potential feasible ways to reduce carbon emissions to promote the realization of carbon neutrality as soon as possible. Therefore, the main implications based on the empirical results are as follows. Throughout the research, the results demonstrated that political mobility has significant effects on the reductions of carbon dioxide emissions in the context of local promotion. The target of carbon neutrality indicated that in the long run the government and local officials will face continuous pressures, so their behavior should no longer be ignored. First, the Chinese central government should pay more attention to the environmental performance of local officials through the appraisal system. GDP can no longer be the most important indicator of political mobility. Second, the behavior of local officials should also be of concern. The promotion path of provincial leaders influences their political performance. By optimizing the appraisal system and rationalizing

how assessment proportions are weighted, local officials can be made to focus on carbon emissions reduction through political mobility. Finally, environmental protection indicators should become the priority during the assessment of government officials.

Furthermore, the carbon emissions trading system is an effective policy tool to reduce carbon intensity and promote carbon neutrality. Several policy implications and suggestions based on this policy can also be drawn. First, due to China's outdated development mode and its secondary industries, which are important causes of its recent energy consumption and carbon emission increases, it is necessary for China to maintain energy conservation and emissions reduction to achieve the goal of carbon neutrality. Second, with the implementation of the carbon emissions trading system, enterprises should develop the new industrialization path and accelerate the transformation of the industrial structure in order to improve the utilization efficiency of energy based on this new type of market-orientated policy emissions reduction policy. Third, by vigorously developing technology-intensive industries to replace pollution-intensive industries and increasing reliance on large-scale and advanced industrial enterprises, the utilization rate of resources can be improved. In short, the carbon trading system and other market-oriented governance tools can serve an important role in dealing with carbon emissions challenges in achieving carbon neutrality. Moreover, the application of advanced technology to secondary industries and emission reduction targets is an additional and more effective way to fundamentally reduce carbon emissions. In other words, the realization of carbon neutrality needs diversified paths, as well as suitable incentive mechanisms.

However, this paper also has some limitations. Owing to limited data availability, this study used provincial panel data to evaluate the effect of carbon trading on emissions reduction. However, there may have been other factors affecting carbon emissions. Future researchers could test other factors based on this study's findings. Carbon emissions trading in a variety of cities should also be investigated to assess its reduction effectiveness. Additionally, more impact variables should be used to establish a larger and more comprehensive database. These approaches could be helpful in obtaining more significant results.

**Author Contributions:** Conceptualization, Z.L.; methodology, Z.L. and B.L.; formal analysis, B.L.; writing—original draft preparation, B.L.; writing—review and editing, Z.L.; supervision, Z.L.; funding acquisition, Z.L. All authors have read and agreed to the published version of the manuscript.

**Funding:** This research has been supported by the National Natural Science Foundation of China (NO. 71974057) and the 'Shu Guang' project (Grant No. 21SG49) supported by the Shanghai Municipal Education Commission and the Shanghai Education Development Foundation.

**Data Availability Statement:** The datasets generated and/or analyzed during the current study are available from the corresponding author on reasonable request.

**Conflicts of Interest:** The authors declare that they have no known competing financial interest or personal relationships that could have appeared to influence the work reported in this paper.

## Abbreviations

CEAD, China Emission Accounts & Datasets; DID, difference-in-differences model; VIF, variance inflation factor.

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
