# Peer review of "Understanding Carbon Emissions Reduction in China: Perspectives of Political Mobility"

_land, doi:10.3390/land12040903_

Round 1
Reviewer 1 Report
1- The author has framed the good idea and report the objectives of the study, but it is important to mention the contribution of the study. Making your research appealing to a broad audience is an important goal of this journal.
2- Related to my previous comment, please highlight whether the proposed econometric approach is best one matching with the economic question addressed in this paper. try to put out this issue in abstract as well as in the introduction.
3- Simple economic intuitions to explain the mechanisms that lead to the findings: Please make sure that the main finding is robust in explaining the mechanism of the impact with unambiguous supporting evidence.
4- Please provide as much practical insights/implications as possible that can attract a general reader to appreciate your finding and therefore can increase the citation of your work when published.
5- Forward-looking statements to motivate a comprehensive reading: The abstract needs to contain the key economic arguments and channels in a logical sequence. Also please try to make your abstract more accessible in a non-technical language.
6- Please revisit your title to make it self-explanatory and intuitively-appealing, reflecting the main idea of the paper in a concise and informative manner.
7- I read the paper carefully, I saw some language issues in the manuscript. Therefore, I recommend to proof read this paper from an English language expert.
Reviewer 2 Report
Thank you very much for your article “Impact of China’s Carbon Trading System on Carbon Emis- 2 sions Reduction: A Perspective of Political Mobility”, that covers one of the most significant issues nowadays – involvement of the authorities in solving environmental problems.
Nevertheless, there are some weaknesses in the study and proposals for the improvement:
1. Chapter 3 should present materials and methods (not research design) according to the recommended structure of the journal, as well as stated structure of scientific articles. The name of the Chapter should be changed.
2. It is necessary to change the numbering of formulas according to the journal template, for example, (3-1) should be changed on (1) on page 6.
3. The variables in formula 3-2 should be explained immediately after the formula or before it.
5. iIt will be better to move "4.1. Descriptive statistics" to the third chapter "Materials and Methods".
Overall, the article has enough validated results, the style of presentation is understandable, the results of the study are interpreted. Therefore, after the revisions, the article can be accepted, if the study is appropriate for the journal and chosen special issue.
Reviewer 3 Report
First of all, I would like to congratulate the authors for the effort made. Secondly, I believe that the selected topic may be of interest for a possible publication if the indicated changes are made. The authors analyze a topic of great interest today such as the impact of political mobility on China's carbon emissions trading policy. The work may not have great international interest, however this type of work may have an impact in their territorial areas and constitute one more tool to establish future decisions on climate action. One of the elements that authors must include is a Discussion section where the results are analyzed and compared with other works. Finally, before its possible acceptance, the authors must make the following changes to improve the work and also have a greater international interest.
a-) Introduction: The authors should improve it so that this section is of greater interest. Almost all of the references provided refer to China. As it is an international journal, I believe that the introduction should introduce works that address issues on climate change mitigation in other places (Europe, the United States, etc.), in addition, it is also important to break down the main sectors that have the highest emissions, such as transport. In this way it will be of greater interest to readers. Any references of interest to include:
1-) https://www.mdpi.com/2071-1050/13/4/1795
2-) https://www.pnas.org/doi/abs/10.1073/pnas.94.1.175
3-) https://www.sciencedirect.com/science/article/pii/S0921800911000814
4-) https://link.springer.com/article/10.1023/A:1011188401445
5-) https://www.nature.com/articles/ncomms8714
b-) Literature review: It is well posed and the hypotheses are of interest. However, in section 2.1 I think they could expand with regulatory elements at the international level.
c-) Research design: The section is well developed. I recommend calling this section Materials and methods to respect the traditional structures of the articles.
c-) Results: The results are well developed and interesting. In addition, they meet the objectives defined at work.
d-) Discussion: The authors have not developed a discussion section, therefore they must include this section, which is essential for an international publication.
e-) Conclusions: The conclusions are of interest, they also address the political implications. However, I think that they can be even more enriching if they also comment on the importance of international agreements such as the Paris Protocol of 2015, a document that the authors do not cite throughout the work.
Round 2
Reviewer 1 Report
The author has addressed all the comments
Reviewer 3 Report
Thank you very much for making the changes you suggested. Congratulations for the effort and work done.